# Live Streaming vs. Pre-Recorded Training during the COVID-19 Pandemic in Italian Rhythmic Gymnastics

**DOI:** 10.3390/ijerph192416441

**Published:** 2022-12-08

**Authors:** Ludovica Cardinali, Davide Curzi, Emanuela Maccarani, Lavinia Falcioni, Matteo Campanella, Dafne Ferrari, Claudia Maulini, Maria Chiara Gallotta, Giovanna Zimatore, Carlo Baldari, Laura Guidetti

**Affiliations:** 1Department of Movement, Human and Health Sciences, University of Rome “Foro Italico”, 00135 Rome, Italy; 2Department Unicusano, University “Niccolò Cusano”, 00166 Rome, Italy; 3Italian Gymnastics Federation “FGI”, 00196 Rome, Italy; 4Faculty of Psychology, eCampus University, 22060 Novedrate, Italy; 5Department of Movement Sciences and Wellbeing, University of Naples Parthenope, 80133 Napoli, Italy; 6Department of Physiology and Pharmacology “Vittorio Erspamer”, Sapienza University of Rome, 00185 Rome, Italy; 7Department of Theoretical and Applied Sciences, eCampus University, 22060 Novedrate, Italy

**Keywords:** online training, live streaming, pre-recorded, COVID-19, rhythmic gymnastics, mobile health

## Abstract

The SARS-CoV-2 outbreak led to an inevitable and drastic transition to online training systems. This study aimed to analyze the differences between live-streaming and pre-recorded training modalities in rhythmic gymnastics (RG) with coaches and gymnasts of different technical levels. A total of 238 coaches and 277 athletes affiliated with the Italian Gymnastics Federation (FGI) participated in the study. The data collection tool was a self-collected questionnaire structured in three sections: participant’s information, characteristics of live-streaming training, and characteristics of pre-recorded training. A 2 × 2 ANOVA was used for each numeric variable. A Pearson’s chi-squared test was used for each categorical variable. For the athletes, training frequency, motivation, and efficacy were significantly higher (*p* < 0.05) with live streaming (3.7 ± 1.5 day/week, 3.8 ± 0.9 score and 3.8 ± 0.8 score, respectively) than with a pre-recorded modality (2.2 ± 1.7 day/week, 3.1 ± 1.2 score and 3.7 ± 0.9 score, respectively), while for coaches, significant differences (*p* < 0.005) were found between the two modalities only for training frequency (live streaming, 3.6 ± 1.8 days/week vs. pre-recorded, 2.1 ± 1.7 days/week). The adherence (number of No:Yes) was significantly higher (*p* < 0.05) for the live-streaming modality than for the pre-recorded modality in gold athletes (1:74 vs. 14:61, respectively), silver athletes (12:190 vs. 28:174, respectively), and gold coaches (3:63 vs. 11:55, respectively), but it was not for silver coaches. Physical preparation was performed significantly (*p* < 0.005) more often (number of No:Yes) for live streaming than for the pre-recorded modality by gold athletes (9:66 vs. 34:41, respectively), silver athletes (25:177 vs. 77:125, respectively), gold coaches (8:58 vs. 37:29), and silver coaches (33:139 vs. 85:87, respectively). Free body technical preparation was performed significantly (*p* < 0.005) more often (number of No:Yes) for live streaming than for the pre-recorded modality by gold athletes (15:60 vs. 39:36, respectively), silver athletes (84:118 vs. 121:81, respectively), gold coaches (10:56 vs. 38:28), and silver coaches (60:112 vs. 105:67, respectively), while no differences were found for silver athletes’ and coaches’ technical preparations for apparatus training between the two modalities. In conclusion, live streaming had greater positive effects on RG training during home confinement. However, pre-recorded training could be more effective for some types of training, depending on the technical level of the athletes.

## 1. Introduction

On 11 March 2020, the SARS-CoV-2 outbreak was declared a global pandemic by the World Health Organization [1]. To avoid mass contagion, all countries implemented protective measures aimed at physical distancing, isolation/quarantine, and, in extreme cases, national lockdown [2]. The Italian government established a series of strict containment measures which had enormous consequences for public life, in general, and for the field of sport, these measures modified the lives, planning, and routines of athletes and coaches [3]. Although the authorities announced that exercise performed alone in outdoor public places was still permitted, sports clubs, fitness centers, swimming pools, and other sports and leisure facilities were closed [4]. In addition, all ongoing championships were suspended, and the major national and international events were postponed (e.g., the 2020 summer Olympics games in Tokyo) [5]. All these measures resulted in several direct consequences for both elite and amateur sports such as the absence of organized training and competition, limited, and mostly denied, access to effective training environments, a lack of physical communication between athletes and coaches, inadequate training conditions, and financial losses [6,7]. During the COVID-19 period of home confinement, athletes were likely exposed to some level of detraining (e.g., partial or complete loss of training-induced morphological and physiological adaptations) as a consequence of the reduction or cessation of training stimuli [5,6,8]. Furthermore, a recent study by Di Fronso et al. [9] showed that the pandemic scenario had a strong detrimental impact on athletes’ perceived stress and psychobiosocial states, with a significant stress increase compared to such situations before the COVID-19 pandemic.

To compensate for the absence of sports facilities and to maintain athletes’ physical and technical conditions, some sports clubs used digital technology to provide athletes with home-based training programs [5]. During the COVID 19 pandemic, digital technologies became a popular tool for the promotion of physical activity (e.g., wearables and the delivery of physical exercise sessions via online classes or smartphone apps), and this trend was strongly sustained by the OMS through the promotion of mobile health (mHealth) strategies to increase physical activity [10]. According to the ACSM Worldwide Survey of Fitness Trends for 2021 [11], online training, which included synchronized live-streamed lessons or asynchronized pre-recorded lessons, increased during the COVID-19 pandemic, becoming the number one trend. The synchronous format was based on real-time interpersonal communication, the use of natural language, and immediate feedback [12], offering the possibility of creating shared work groups, monitoring the progress of training, and directly creating interactions between athletes and coaches. Conversely, an asynchronous modality is temporally and geographically independent, and it is defined as more individually based, self-paced, and less instructor-dependent [12]. It consists of lessons prepared by a coach or retrieved from the internet and sent to athletes to use in complete autonomy.

Many forms of online training were used during the home confinement period, with some positive outcomes reported regarding the coach–athlete relationship [13]. A study by Piatti et al. [14] showed that professional Italian volleyball players performed submaximal and low-intensity exercises in complete autonomy alone or assisted via webcam in real-time by coaches to avoid the total or partial loss of the adaptations achieved through season training. Qualitative research applied to boxing clubs, boxers, and boxing coaches in Norway showed that many clubs had incorporated online training systems, both live-streamed using digital platforms such as Zoom and Microsoft Teams and pre-recorded videos—in some cases, exclusively for members of a specific club, and in others, open to anyone interested in participating [15].

Changes in training delivery systems occurred in different sports at all levels, including rhythmic gymnastics (RG). RG is an artistic sport characterized by different levels of competition based on ages and level of performance. It is practiced both individually and in groups and is performed with technical apparatuses (ropes, hoops, balls, clubs, and ribbons), and it requires high levels of strength, flexibility, and movement precision [16].

Only one study described what was used during home confinement in a sample of RG coaches [17]. However, it would appear appropriate to better understand how gymnasts and coaches used online training systems (live streaming and/or pre-recorded) and their characteristics to determine how to provide exercise training and coaching efficiently and safely through this method. More specifically, the present study examined the experience of both athletes and coaches with different levels of experience to provide a double perspective of training during confinement. Indeed, a literature study has shown that the level of technical experience can determine the different effects of the training variables [18]. Moreover, understanding which online modality determines greater adherence could be important for future situations. Several studies have highlighted the importance of factors such as motivation, exercise efficacy, and training frequency on adherence and participation in exercise training programs [19,20,21]. Therefore, this research aimed to analyze the differences between live streaming and a pre-recorded online modality in both coaches and gymnasts of different technical levels for: (1) adherence and frequency (times per week) of training; (2) motivation and efficacy of training; and (3) type of training performed.

## 2. Materials and Methods

### 2.1. Participants

The sample consisted of 238 RG coaches (aged 33.60 ± 10.32 years old; frequency of training before COVID-19 was 4.75 ± 1.15 times per week) from different Italian regions (north: 51.3%; center: 41.2%; and south: 7.6%) and 277 RG athletes (aged 13.57 ± 2.81 years old; frequency of training before COVID-19 was 4.19 ± 1.14) from different Italian regions (north: 65.3%; center: 29.6%; and south: 5.1%). The technical levels of the coaches (considering the competition level of the gymnasts they trained) and the athletes depended on the type of competition in which they participated, and it was divided into two categories proposed by FGI: (1) gold: high-level competitions and (2) silver: medium-level competitions. The inclusion criteria for this study were: (i) affiliation with the Italian Gymnastics Federation (FGI) and (ii) participation in FGI-silver or FGI-gold competitions.

### 2.2. Data Collection

A questionnaire was specifically created for the present study by RG training experts, including the Italian national technical director of RG and university professors in sports exercise. The questionnaire was composed of three sections: (i) participant’s information (sports role, age, geographical region, and technical level) and frequency of training before the COVID-19 pandemic, (ii) characteristics of live-streaming training, and (iii) characteristics of pre-recorded training.

The questionnaire collected self-reported information about the following variables: (1) adherence to online training: participants indicated if they continued training through live-streamed and/or pre-recorded video lessons during home confinement by answering “yes” or “not”; (2) frequency of online training: participants indicated weekly online training frequency ranging from 1 to 7 days per week; and (3) type of training performed: physical preparation, apparatus technical work, and free-body technical work. Participants indicated the type of training performed, choosing between physical preparation aimed at developing organic-muscular abilities such as flexibility, strength, speed, and endurance using free-weights, resistance bands, and non-coded tools and technical preparation with rhythmic gymnastics small apparatuses aimed at learning the technical elements such as apparatus difficulty (AD); small and large throws; free-body technical preparation aimed at learning jumps, balancing, and rotation elements from the difficulty tables in the Code of Points; acrobatic elements such as cartwheels, forwards and backward walkovers, forwards and backward rolls; and classical dance elements. The two additional variables were: (4) motivation of online training: participants indicated how motivating the training was, with possible answers ranging from 1, which was very negative, to 5, which was very positive; and (5) efficacy of online training: participants indicated how effective the training was, with possible answers ranging from 1, which was completely ineffective, to 5, which was completely effective.

The questionnaire was administered to RG coaches and gymnasts during the last phase of the Italian lockdown period (25 April 2020 to 25 June 2020). It was based on an anonymous self-administered web survey via Google forms. The distribution of the questionnaire was coordinated by the national technical director of the Italian Gymnastics Federation (FGI) through the regional gymnastics committees. Before the beginning of the questionnaire, all participants received introductory comments concerning the rationale for the study, the use of the data, and the topics of investigation. Each section of the questionnaire had to be completed for it to be accepted. The questionnaire compilation for under-18 gymnasts was supervised by a parent.

### 2.3. Data Analysis

Statistical analysis was performed separately for athletes and coaches. A 2 × 2 ANOVA was performed for each numerical variable (frequency, motivation, and efficacy of training), with technical level (gold vs. silver) as the between-participants factor and online training modality (live streaming vs. pre-recorded training) as the within-participants factor. The effect size was also calculated using Cohen’s definition of small, medium, and large effect size as partial ƞ^2^ = 0.01, 0.06, and 0.14, respectively [22]. Significant interactions were further analyzed using t-test comparisons. Pearson’s chi-square test was used for categorical variables (adherence and type of training). Statistical significance was defined as *p* ≤ 0.05. Statistical analysis was performed using IBM SPSS Statistics for Windows, version 23.0 (Armonk, NY, USA: IBM Corp).

## 3. Results

All results were expressed as means ± standard deviations. The ANOVA results are reported in Table 1.

### 3.1. Athletes’ Results

The main effect of the training modality revealed that training frequency (live streaming: 3.7 ± 1.5 day/week vs. pre-recorded: 2.2 ± 1.7 day/week), training motivation (live streaming: 3.8 ± 0.9 score vs. pre-recorded: 3.1 ± 1.2 score), and training efficacy (live streaming: 3.8 ± 0.8 score vs. pre-recorded: 3.7 ± 0.9 score) were significantly higher for live streaming than for pre-recorded training. The main effect of the technical level revealed that gold athletes had higher training frequencies than silver athletes (3.6 ± 2.0 days/week vs. 2.7 ± 1.7 days/week, respectively). Finally, training modality–technical level interaction revealed that gold athletes’ training frequencies were significantly higher than those of silver athletes in the live-streaming training modality (Figure 1A).

Adherence to online training was significantly higher for the live-streaming modality than for the pre-recorded modality for both gold and silver athletes (Table 2). Regarding the type of training performed during the confinement period, physical preparation and free body technical preparation were significantly more often performed for the live-streaming modality than for the pre-recorded modality for both gold and silver athletes, while apparatus technical preparation was significantly more often performed for the live-streaming modality than for the pre-recorded modality for gold athletes but not for silver athletes.

### 3.2. Coaches’ Results

The main effect of the training modality revealed that training frequency was significantly higher for live streaming than for pre-recorded training (3.6 ± 1.8 days/week vs. 2.1 ± 1.7 days/week, respectively). The main effect of technical level revealed that gold coaches had higher training frequencies than silver coaches (3.3 ± 2.1 days/week vs. 2.6 ± 1.8 days/week, respectively). Finally, the training modality–technical level interaction revealed that gold coaches’ training frequencies were significantly higher than those of silver coaches for the live-streaming training modality (Figure 1D).

Adherence to online training was significantly higher for the live-streaming modality than for the pre-recorded modality for gold coaches but not for silver coaches (Table 2). Regarding the type of training performed, physical preparation and free body technical preparation were significantly more often performed in the live-streaming modality than in the pre-recorded modality for both gold and silver coaches, while no significant difference was found in the apparatus technical preparation between the live-streaming modality and the pre-recorded modality for both gold and silver coaches.

## 4. Discussion

The temporary closure of sports clubs around the world due to the spread of SARS-CoV-2 led to an inevitable and drastic transition towards online training, resulting in a demand for in-depth research on effective ways to conduct online training for athletes.

The results of this study showed that the live-streaming modality had greater positive effects on rhythmic gymnastics training during the period of home confinement than the pre-recorded modality for both athletes and coaches.

Many forms of online training were used during the COVID-19 home confinement period [4,14,15,23]. To our knowledge, this is the first study aimed at investigating the differences between the live-streamed and pre-recorded online training modalities for RG athletes and coaches of different technical levels during the first COVID-19 national lockdown in Italy.

The results of our study showed that for all athletes, online training adherence was higher with live streaming than with the pre-recorded modality. This result is in line with that of Bobo-Arce et al. [17], who reported a greater use of the real-time training system by RG coaches with their gymnasts. However, the result was not obvious since there were conditions under which live streaming was not possible. An inability to attend the live lesson at the scheduled time, a lack of appropriate support to young gymnasts from parents engaged in work activities, an absence of adequate home spaces, a scarce fluency of interaction which determines lower attention [24], and technical difficulties such as computer or internet crashes were some of the problems associated with live-streaming lessons [12,25]. In these cases, the pre-recorded modality was preferable, allowing training at more convenient times and with the support of parents. In addition, the pre-recorded modality has functions such as ‘play’, ‘forward’, ‘rewind’, and ‘re-watch’, which offers athletes the convenience of watching and performing exercises at their speed [26]. However, even in sports clubs where both training modalities were proposed, there was a preference for live streaming. For coaches, the adherence was higher with the live-streaming modality than the pre-recorded modality only for the gold—but not silver—level athletes. This result is in contrast to that of Bobo-Arce et al. [17], who found that all coaches, regardless of the gymnasts’ levels, used real-time video conferencing to conduct training sessions. Our finding can be explained by a greater need at the gold level to correct the details of the exercises and, therefore, to see and examine the athletes’ performance directly. The use of both training modalities at the silver level could be due to the high importance of the repetition of the routine and technical elements for silver athletes.

Both athletes and coaches reported reduced training frequency, from four sessions per week before COVID-19 to three and two sessions per week with live streaming and the pre-recorded modality, respectively, during the lockdown period. Our result is in line with that of a previous study by Washif et al. [27], who reported a marked reduction in athletes’ training frequencies during the COVID-19 lockdown period. However, in both athletes and coaches, the frequency of training per week was higher with the live-streaming modality than with the pre-recorded modality. This result may be because the live-streaming modality allowed for group interactions, as well as for greater control of training, and therefore it was more frequently used by coaches and athletes. Our results also showed that gold athletes and coaches had higher training frequencies than silver only with the live-streaming modality and not with the pre-recorded modality. The gold level requires more control by the coach than the silver level due to the high level of performance, which is better guaranteed by live streaming than by the pre-recorded modality.

Rhythmic gymnastics is an artistic sport that requires a high degree of physical, technical, and psychological skill aimed at obtaining a perfect execution of body movements with different types of equipment (balls, ribbons, hoops, clubs, and ropes) [28]. Bobo-Arce et al. [17] showed that rhythmic gymnasts’ training during home confinement was mainly focused on physical fitness maintenance and improvement, while technical contents were scarcely included. Our results showed that for both athletes and coaches, physical and free body technical preparations were more often performed with live streaming than with the pre-recorded modality for both the gold and silver levels. This result may be explained by the increased need to directly supervise athletes due to potential risks of injury related to incorrect execution of the movements. Physical preparation, aimed at developing flexibility, strength, and aerobic capacity, included activities such as stretching and power exercises that require a large volume of training performed with the use of free weights, resistance bands, and non-coded tools [29,30]. Similarly, free body technical preparation aimed at learning body difficulties and acrobatic elements requires correct body positioning for safe completion. When using pre-recorded training, the ability to provide direct supervision over athletes is not available. Therefore, an athlete could perform incorrect movements, improperly load an exercise, or misunderstand instructions, potentially increasing the risk of injury. On the other hand, with the live-streaming modality, coaches can directly supervise athletes who can better understand corrections. In addition, the live-streaming modality allows for individualized training load and recovery, which are the main causes of gymnasts’ injuries [31,32]. Differently, the apparatus technical preparation was more often performed with live streaming compared to the pre-recorded modality only by gold athletes, while no differences were found between the two modalities with coaches. This result suggests that the apparatus technical training, which requires a significant number of repetitions of technical elements until perfect execution is achieved [30,33,34,35], is more easily performed through a pre-recorded modality that allows athletes to re-watch and repeat technical elements. For gold athletes, where higher-level apparatus technical elements require more detailed explanations from coaches, the live-streaming modality was preferred.

Motivation is an important factor in adopting and maintaining physical exercise [19]. Some studies have shown that maintaining an athlete’s motivation and well-being during the COVID-19 period of home confinement was a critical challenge [36,37]. Therefore, online tools could represent a valid instrument for facilitating the practice of physical activity at home. The results of our study showed that both gold and silver athletes were more motivated to use live streaming than they were to use the pre-recorded training modality. This finding is in accordance with the research of Newbold et al. [23], who showed that live-streaming classes offered a way to stay connected with people during the lockdown and to continue with exercise classes they would previously have attended in person, providing a sense of community. These results are in line with a study by Bonavolontà et al. [38] on the level of enjoyment of home-based training for 140 children and adolescents from 23 different Apulian Tennis Clubs, which showed that those who received online support and guidance from their instructors showed significantly higher average enjoyment levels than the group who had no such indication. Moreover, Lautenbach et al. [36] reported that the coping strategy most used by athletes during lockdown was related to online training and exercise, using tools and software to maintain social contact. Contrary to our expectations, no differences were found between the live and pre-recorded modalities in coaches’ training motivation. This result might be explained by a greater variety of exercises being available through pre-recorded lessons retrieved from the internet, which led coaches to develop new effective and motivating exercise routines. Training efficacy measures the impact of online training on the athletes’ knowledge, skills, and performance, and its evaluation is important to determine whether the training benefits athletes’ skills and performances. In our study, the athletes reported significantly higher training efficacy scores for live streaming than for the pre-recorded modality, while for coaches, no significant differences were found between the two modalities. The live-streaming modality showed greater benefits for the athletes, likely due to the greater interaction with their coaches, who considered both modalities effective for their athletes’ preparation. Furthermore, this study has demonstrated the importance of working with an interdisciplinary team [39,40].

## 5. Conclusions

The recent development of mobile health has led to the increasing use of mobile and multimedia telecommunications for the promotion of health and well-being. In this context, the results of our study provide useful information on how to effectively use digital technology to induce and maintain the psychophysical well-being of athletes, both professional and amateur, with possible implications applicable to the general population, for example, the identification of a telecommunication modality that allows wide involvement and greater motivation to engage in physical exercise.

The results of our study showed that live streaming was the most used modality for distance training for both RG athletes and coaches, and it was the most motivating and effective modality for athletes. However, the determinants for the development of online distance training vary depending on the technical level of the athlete. Combining live streaming with pre-recorded sessions within home-based athlete training programs based on the technical level and type of preparation can be effective. In addition, variables such as coaches’ background, previous experience, and age should be considered as they can affect the athlete’s ability to engage in online training systems and understand content through the live-streaming and pre-recorded approaches. Furthermore, the present study may serve as a stimulus for sports clubs to further reflect on how to effectively use digital technology to induce positive effects on training, as well as to orient and plan similar future situations.

## Figures and Tables

**Figure 1 ijerph-19-16441-f001:**
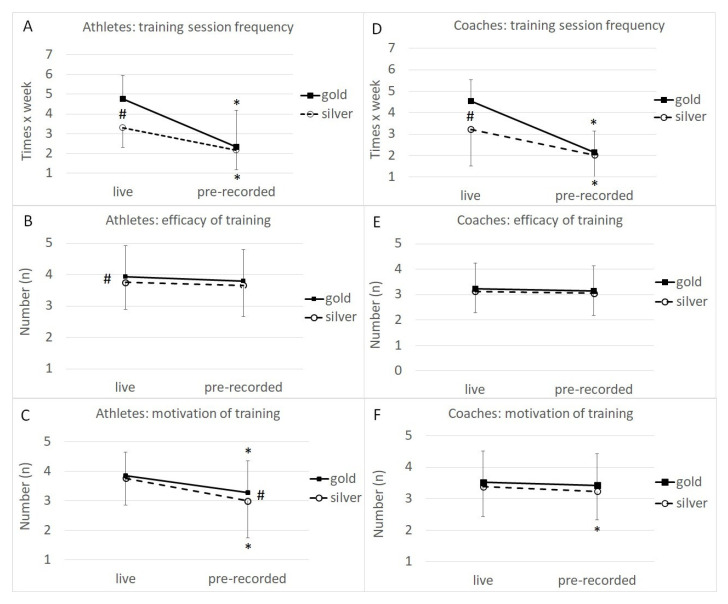
Effect of training modality (live streaming vs. pre-recorded) for gold (dark line) and silver (dashed line) technical level athletes (**A**–**C**) and coaches (**D**–**F**) on training session frequency (**A**,**D**), training efficacy (**B**,**E**), and training motivation (**C**,**F**). * *p* < 0.05 vs. live; # *p* < 0.05 gold vs. silver.

**Table 1 ijerph-19-16441-t001:** ANOVA results for athletes and coaches for the numeric variables of training (frequency, motivation, and efficacy), with technical level (gold vs. silver) as the between-participants factor and online training modality (live streaming vs. pre-recorded) as the within-participants factor.

	Variable	Factors	df	F	*p*	Partial ƞ^2^
Athletes						
	Frequency	Modality **	1	142.259	<0.001	0.341
		Technical level **	1	30.243	<0.001	0.099
		Modality × technical level **	1	18.984	<0.001	0.065
		Error	275			
	Motivation	Modality **	1	39.219	<0.001	0.151
		Technical level	1	2.21	0.139	0.010
		Modality × technical level	1	0.873	0.351	0.004
		Error	220			
	Efficacy	Modality *	1	4.66	<0.05	0.021
		Technical level	1	1.786	0.183	0.008
		Modality × technical level	1	0.103	0.748	0.000
		Error	220			
Coaches						
	Frequency	Modality **	1	138.429	<0.001	0.37
		Technical level **	1	13.323	<0.001	0.053
		Modality × technical level **	1	14.983	<0.001	0.06
		Error	236			
	Motivation	Modality	1	3.313	0.07	0.018
		Technical level	1	1.634	0.203	0.009
		Modality × technical level	1	0.16	0.689	0.001
		Error	183			
	Efficacy	Modality	1	1.974	0.162	0.011
		Technical level	1	0.591	0.443	0.003
		Modality × technical level	1	0.03	0.864	0
		Error	278			

** *p* < 0.001 and * *p* < 0.05.

**Table 2 ijerph-19-16441-t002:** Athletes’ and coaches’ answers on training adherence and type of training performed (physical preparation, apparatus technical preparation, and free body technical preparation).

		Gold (n = 75)		Statistics			Silver (n = 202)		Statistics		
Variable		Live	Pre-Recorded	c^2^	*df*	*p*	Live	Pre-Recorded	c^2^	*df*	*p*
Athletes											
	adherence, n No:Yes	1:74	14:61 **	12.5	1	0.001	12:190	28:174 *	7.10	1	0.012
	physical preparation, n No:Yes	9:66	34:41 **	20.4	1	0.000	25:177	77:125 **	35.46	1	0.000
	apparatus technical preparation, n No:Yes	10:65	32:43 **	16.0	1	0.000	67:135	80:122	1.80	1	0.215
	free body technical preparation, n No:Yes	15:60	39:36 **	16.7	1	0.000	84:118	121:81 **	13.55	1	0.000
		Gold (n = 66)		Statistics			Silver (n = 172)		Statistics		
Variable		Live	Pre-recorded	c^2^	*df*	*p*	Live	Pre-recorded	c^2^	*df*	*p*
Coaches											
	adherence, n No:Yes	3:63	11:55 *	5.1	1	0.045	17:155	26:146	2.15	1	0.192
	physical preparation, n No:Yes	8:58	37:29 **	28.4	1	0.000	33:139	85:87 **	34.88	1	0.000
	apparatus technical preparation, n No:Yes	17:49	24:42	1.7	1	0.259	55:117	49:123	1.73	1	0.259
	free body technical preparation, n No:Yes	10:56	38:28 **	25.7	1	0.000	60:112	105:67 **	23.58	1	0.000

** *p* < 0.001 and * *p* < 0.05.

## Data Availability

All relevant data are within the manuscript.

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
