# Peer review of "Live Streaming vs. Pre-Recorded Training during the COVID-19 Pandemic in Italian Rhythmic Gymnastics"

_ijerph, 2022, doi:10.3390/ijerph192416441_

Round 1

Reviewer 1 Report

Dear author,

Your work is very interesting and also very relevant to the current times. I would like to congratulate you for what ypiu have done, and at the same time, suggest some details that could help to improve the quality of your work a little bit more.

Abstract:

Data analysis is not mentioned. I suggest specifying that you used a 2x2 ANOVA. 

Introduction:

This is well constructed with a wide amount of bibliographic references.

Until line 89 the rationale is clear in terms of why and online training is now a trend and what are the offers for it. However, in the paragraph between lines 90 to 92 it conect with the lack of studies on this matter in the context of sports. This paragraph is better placed at the end to connect with the aim of the study. When I read this, I think the introduction is ending. I suggest moving o erase the paragraph and replacing it with a simple connection to the paragraph on the line 93.

Also, according to the aim of the study, the rationale of the introduction only justify the goal about to analyse both coaches and gymnasts the difference between live streaming and pre recorder online modality. However, the introduction does not explain why is relevant to study the adherence, frequency, motivation and efficacy. I would like to read some evidence that could explain why this is relevant for the study.

Materials and Methods

In data collection it is mentioned the use of a specific questionnaire. Is not clear if the questionnaire was created for this study or to study RG in general. If it is the first, it is neccesary to reference it . 

If it is the second, please ammend the writing and specify in line 115 that "A questionnaire was specifically created for the present study by..."

Data Analysis

No suggestion here.

Results

Table 1: regarding p-value, I am assuming that is significant at 0.005. I suggest the use of a * on the significant values and specify that under the table ( *= Significant p-value or Significant p-value at <0.005)

Table 2: the same as table 1

Discussion

Nos suggestions here

Conclusions

Nos suggestions here

Once again, this is very good work, and I hope that my suggestions help to improve a little.

Reviewer 2 Report

This manuscript entitled “live streaming vs pre-recorded training during COVID-19 pandemic in Italian rhythmic gymnastics.

It is important to document the changes of training programs during the pandemic.

Especially, sports and physical training programs have been negatively hit by the pandemic.

The relevant organizations appeared to introduce new delivery system through online platforms. In this regard, this manuscript is very interesting.

If can, the authors should describe more the process and outcomes in detail.

I have some comments.

1.     Literature review

The section of literature review is somewhat well written or is not presented. Please describe the definition of important variables in methods and let readers understand the concepts tested in this paper.

2.     Sampling and the respondents’ information

More specific description about sampling technique, tested training system would be helpful for understanding the context of this study. Moreover, the specific information about the respondents are important.

3.     The discussion provides good information.

Round 2

Reviewer 2 Report

Thank you for revision. I recommend this manuscript for publication.

Author Response

Thanks